# MFS1, a Pleiotropic Transporter in Dermatophytes That Plays a Key Role in Their Intrinsic Resistance to Chloramphenicol and Fluconazole

**DOI:** 10.3390/jof7070542

**Published:** 2021-07-07

**Authors:** Tsuyoshi Yamada, Takashi Yaguchi, Karine Salamin, Emmanuella Guenova, Marc Feuermann, Michel Monod

**Affiliations:** 1Teikyo University Institute of Medical Mycology, Tokyo 192-0395, Japan; 2Asia International Institute of Infectious Disease Control, Teikyo University, Tokyo 173-0003, Japan; 3Medical Mycology Research Center, Chiba University, Chiba 260-8673, Japan; yaguchi@chiba-u.jp; 4Department of Dermatology, Centre Hospitalier Universitaire Vaudois, 1011 Lausanne, Switzerland; Karine.salamin@chuv.ch (K.S.); Emmanuella.Guenova@chuv.ch (E.G.); michel.monod@chuv.ch (M.M.); 5Faculty of Biology and Medicine, University of Lausanne, 1015 Lausanne, Switzerland; 6Swiss-Prot Group, SIB Swiss Institute of Bioinformatics, 1205 Geneva, Switzerland; Marc.Feuermann@sib.swiss

**Keywords:** dermatophytes, *Trichophyton benhamiae*, chloramphenicol, cycloheximide, intrinsic resistance, azole resistance, MFS transporters, ABC transporters, dermatology

## Abstract

A recently identified *Trichophyton rubrum* major facilitator superfamily (MFS)-type transporter (TruMFS1) has been shown to give resistance to azole compounds and cycloheximide (CYH) when overexpressed in *Saccharomyces cerevisiae*. We investigated the roles of MFS1 in the intrinsic resistance of dermatophytes to CYH and chloramphenicol (CHL), which are commonly used to isolate these fungi, and to what extent MFS1 affects the susceptibility to azole antifungals. Susceptibility to antibiotics and azoles was tested in *S. cerevisiae* overexpressing MFS1 and ΔMFS1 mutants of *Trichophyton benhamiae*, a dermatophyte that is closely related to *T. rubrum*. We found that TruMFS1 functions as an efflux pump for CHL in addition to CYH and azoles in *S. cerevisiae*. In contrast, the growth of *T. benhamiae* ΔMFS1 mutants was not reduced in the presence of CYH but was severely impaired in the presence of CHL and thiamphenicol, a CHL analog. The suppression of MFS1 in *T. benhamiae* also increased the sensitivity of the fungus to fluconazole and miconazole. Our experiments revealed a key role of MFS1 in the resistance of dermatophytes to CHL and their high minimum inhibitory concentration for fluconazole. Suppression of MFS1 did not affect the sensitivity to CYH, suggesting that another mechanism was involved in resistance to CYH in dermatophytes.

## 1. Introduction

Dermatophytes are highly specialized keratinophilic filamentous fungi and are the most common pathogenic agents of skin, nail, and hair mycoses [1,2,3]. The isolation of dermatophytes is carried out after the deposition of a dermatological sample on a nutrient agar medium, most often a Sabouraud medium. The growth of dermatophytes is relatively slow in comparison to other fungi and their isolations are regularly complicated by the contamination of the cultures by all kinds of fast-growing molds and bacteria. However, the intrinsic resistance of dermatophytes to cycloheximide (CYH) and chloramphenicol (CHL) has long been exploited in routine diagnostic laboratories for the isolation of these fungi [1,2,3]. Thus, to isolate dermatophytes, two cultures are generally made in parallel, one on Sabouraud medium containing CHL at a concentration of 50–200 µg/mL to inhibit bacteria, and the other on Sabouraud medium containing CHL and CYH at a concentration of 400–500 µg/mL to inhibit the growth of molds. CHL binds to the center of bacterial ribosome peptidyl transferase and prevents bacterial translation by inhibiting the elongation of polypeptides. CHL also affects the function of eukaryotic mitochondrial ribosomes and translation in the mitochondria [4]. CHL can act as an antifungal agent as well [5]. On the other hand, CYH is an inhibitor of the initiation and elongation phases in protein synthesis in eukaryotic cells [6]. CYH blocks the growth of most fungi. In the middle of the 1900s, Whiffen showed that zoopathogenic fungi were 10 to 15 times more resistant to CYH than phytopathogens or saprophytes [7,8].

In *Trichophyton rubrum*, we recently identified an MFS-type transporter, TruMFS1, which is capable of conferring resistance to *Saccharomyces cerevisiae* to five azoles (itraconazole (ITC), voriconazole (VRC), miconazole (MCZ), fluconazole (FLC), and ketoconazole (KTC)) and, in addition, to CYH when its coding cDNA was overexpressed in yeast [9]. Five azole efflux pumps, including four ABC-type transporters (TruMDR1, TruMDR2, TruMDR3, and TruMDR5), and a second MFS-type transporter (TruMFS2) were also identified [9,10,11]. However, none of these transporters were shown to act as an efflux pump in yeast for CYH [9,12].

In this study, we have performed a targeted disruption of the *MFS1* gene in the dermatophyte *Trichophyton benhamiae* to identify the roles of this transporter in the intrinsic resistance of dermatophytes to CYH as well as to azole compounds and antibiotics targeting mitochondrial translation including CHL. Our experiments revealed that MFS1 is a key player in dermatophyte resistance to CHL and the minimal inhibitory concentration (MIC) of FLC. However, the suppression of MFS1 by gene replacement did not affect the sensitivity to CYH, suggesting that another mechanism is involved in resistance to CYH in dermatophytes.

## 2. Materials and Methods

### 2.1. Strains and Growth Media

A wild-type *T. benhamiae* strain CBS112371 [13] and *T. benhamiae Ku70*-lacking mutant ΔTbeKu70-5-1 (TIMM40015; IFM 66349), which was produced from a wild-type strain IHEM20161 (+), were used in this study. *Trichophyton benhamiae* was grown on Sabouraud dextrose agar (SDA) (1% (*w*/*v*) Bacto peptone (BD Bioscience), 2% (*w*/*v*) dextrose, 2% (*w*/*v*) agar), in Sabouraud dextrose broth medium (SDB), as well as on Sabouraud glycerol agar (SGA) medium (1% (*w*/*v*) Bacto peptone, 2% (*w*/*v*) glycerol, 2% (*w*/*v*) agar). Spore production was induced at 28 °C using 1/10 SDA (0.1% (*w*/*v*) Bacto peptone (BD Bioscience), 0.2% (*w*/*v*) dextrose, 2% (*w*/*v*) agar). For *MFS1* (*TbeMFS1*) gene disruption in *T. benhamiae*, *Agrobacterium tumefaciens* EHA105 was maintained as previously described [14].

*Saccharomyces cerevisiae* Y02409 (*MAT*a; *ura3Δ0*; *leu2Δ0*; *his3Δ1*; *met15Δ0*; *YOR153w::kanMX4*) (Euroscarf) was used for overexpression of *T. rubrum* cDNA library. Complete medium for *S. cerevisiae* consisted of 1.0% (*w*/*v*) Bacto yeast extract (BD Bioscience), 2.0% (*w*/*v*) Bacto peptone, and 2.0% (*w*/*v*) dextrose (YPD). YPG was prepared in the same way as YPD with 2.0% (*w*/*v*) galactose instead of dextrose. *S. cerevisiae* synthetic minimal medium (MMD), supplemented with histidine, leucine, methionine, and tryptophan (20 μg/mL), was prepared according to Sherman and Wakem (1991) [15], with 2.0% (*w*/*v*) dextrose as the carbon source. For the expression of genes cloned in pYES2-DEST52 under the control of the *GAL1* promoter, galactose was added instead of dextrose as the carbon source (MMG medium). YPD, YPG, MMD, and MMG plates were made with 2.0% (*w*/*v*) agar. Solid growth medium with glycerol (YCAG) consisted of 0.2% (*w*/*v*) Bacto yeast extract, 0.2% (*w*/*v*) casamino acids (BD Bioscience), 3.0% (*w*/*v*) glycerol, and 2.0% (*w*/*v*) agar. Sterile 80% (*w*/*v*) glycerol was added after the sterilization of the other components (yeast extract, casamino acids, and agar in deionized water) immediately prior to pouring medium into Petri dishes.

All plasmid preparations were made with *E. coli* DH5α or XL-1 Blue.

### 2.2. Chemicals

CHL (Sigma-Aldrich, St. Louis, MI, USA) and thiamphenicol (TAP) (Wako Pure Chemical, Osaka, Japan) were dissolved in ethanol at a concentration of 20 mg/mL, whereas erythromycin (ERY) (Sigma-Aldrich) was dissolved in ethanol at a concentration of 10 mg/mL. Tetracycline (TET) (Sigma-Aldrich) was dissolved in ethanol at a concentration of 5 mg/mL. FLC, ITC, VRC, and MCZ were dissolved in dimethyl sulfoxide (DMSO) to constitute stock solutions of 0.5 mg/mL. Stock solutions were stored at −20 °C until use. Non-impregnated cellulose discs (BioMérieux, Marcy-l’Étoile, France) were used for the diffusion assays.

### 2.3. TruMFS1 Heterologous Expression in S. cerevisiae and CHL Susceptibility Assays

A plasmid for constitutive expression of *MFS1* in *S. cerevisiae* was constructed using the expression vector p426GPD [16] (Table 1). This plasmid harbors the strong constitutive *GPD* promoter upstream of a polynucleotide cloning site. The pYES2-DEST52 plasmid (Invitrogen, Carlsbad, CA, USA) harboring a cDNA fragment encoding TruMFS1 [9], was digested by the restriction enzymes BamHI and NotI. The isolated cDNA fragment was inserted end-to-end into the *S. cerevisiae* expression vector p426GPD digested by the same enzymes. The generated plasmid was designated as pMFS1-2 (Table 1). Nucleotide sequencing of plasmid DNAs was performed by Microsynth (Balgach, Switzerland). All plasmid vectors were prepared with *E. coli* DH5α or XL-1 Blue.

The genetic transformation of *S. cerevisiae* was performed with 1.0 μg of p426GPD or pMFS1-2, using a transformation kit (Invitrogen) according to the manufacturer’s recommendations. The selection of *URA3* transformants was performed using MMD plates and required amino acids.

Yeast transformants were grown to the mid-exponential phase (OD_600_ = 1.0) at 30 °C in MMD liquid medium supplemented with the required amino acids. Each culture was diluted to an optical density at 600 nm of 1.0. An OD value of 1.0 was found to correspond to about 10^7^ colony forming units (CFU)/mL in the yeast suspensions. To test resistance to CHL, ERY, and TET, 10^6^
*S. cerevisiae* transformant cells were seeded to make a lawn on YCAG agar-solidified medium plates. Non-impregnated cellulose discs, 6 mm in diameter, were individually loaded with 20 and 100 µg of antibiotics and then placed at the surface of the medium. The plates were incubated at 30 °C and growth inhibition was observed after 5 days. Alternatively, *S. cerevisiae* transformants were tested for CHL resistance using serial dilution drug susceptibility assays. In these assays, 10 μL of serial dilutions (10^0^ to 10^4^ cells) of the yeast suspension were spotted onto YCAG plates containing a concentration of 100 μg/mL CHL.

### 2.4. Construction of Gene Replacement Vectors for Targeted-Gene Disruption

A *TbeMFS1*-targeting vector, pAg1-*TbeMFS1*/T (Figure 1A, Table 1), was constructed as follows. Approximately 2.0 kb of the upstream and downstream fragments of *TbeMFS1* (GenBank accession no. DAA75241) were amplified from ΔTbeKu70-5-1 total DNA by PCR with the primer pairs AbF150-AbR140 and AbF152-AbR143, respectively. The *E. coli* neomycin phosphotransferase gene (*nptII*) cassette, which is composed of the promoter sequence of *Aspergillus nidulans trpC* gene (*PtrpC*), *nptII,* and the terminator of *Aspergillus fumigatus cgrA* gene (*TcgrA*), was amplified from the plasmid vector pSP72-PcFLP (Table 1) by PCR with the pair of primers PtrpC-F1-TcgrA-R1 (Appendix A). The three amplified fragments were digested with SpeI/ApaI, BamHI/KpnI or ApaI/BamHI, respectively, and cloned into SpeI/KpnI double-digested pAg1 [17], to generate pAg1-*TbeMFS1*/T (Figure 1A, Table 1). A DNA fragment containing the 5′ UTR and ORF of *TbeMFS1* was amplified from Δ TbeKu70-5-1 total DNA by PCR with the pair of primers AbF155-AbR145. The *PtrpC* and *TcgrA* fragments were amplified from the binary vector pAg1-*TruMDR3*/T (Table 1) with primer pairs PtrpC(F)ApaI-PtrpC-hph(R) and hph-TcgrA(F)-TcgrA(R)BamHI, respectively, and the *E. coli* hygromycin B phosphotransferase gene (*hph*) was amplified from the binary vector pAg1-*hph* [14] (Table 1) by PCR with the primer pair PtrpC-hph(F)-hph-TcgrA(R). The three amplified fragments were fused by overlap extension PCR with the PtrpC(F)ApaI-TcgrA(R)BamHI primers, resulting in the generation of the hygromycin B resistance (*hph*) cassette. The amplified upstream and ORF fragment of *TbeMFS1* and the *hph* cassette were digested with SpeI/ApaI or ApaI/BamHI, respectively, and cloned into *Spe*I/*Bam*HI double digested pAg1-*TbeMFS1*/T, resulting in the generation of the pAg1-*TbeMFS1*/C, a complementation vector for the *TbeMFS1* disruptant (Figure 1A, Table 1).

The PCRs were performed using PrimeSTAR HS DNA polymerase (Takara Bio, Kusatsu, Japan). All of the internal ApaI and KpnI sites contained in the amplified fragments were inactivated by overlap extension PCR with pairs of corresponding primers, respectively. The nucleotide sequences of all of the primers used are listed in Appendix A. If necessary, the amplified fragments were gel purified with a QIAEX II gel extraction kit (Qiagen, Hilden, Germany), subcloned into HincII-digested pUC118, and sequenced.

### 2.5. Fungal Genetic Transformation

Genetic transformation of *T. benhamiae* was performed by the *Agrobacterium tumefaciens*-mediated transformation (ATMT) method as described previously [14] with minor modifications. After co-cultivation, nylon membranes were transferred onto SDA medium containing 250 µg/mL G418 or hygromycin B (Wako Pure Chemical, Osaka, Japan) and 200 µg/mL cefotaxime sodium (Sanofi K.K., Tokyo, Japan), overlaid with 10 mL of SDA supplemented with the same concentration of G418 or hygromycin B and cefotaxime sodium and incubated at 28 °C. The plates were further overlaid after 24 h with 10 mL of SDA containing 350 µg/mL G418 or 400 µg/mL hygromycin B and 200 µg/mL cefotaxime sodium and then incubated at 22 to 28 °C for 3 to 4 days, according to the development of colonies on the surface of the plates. The colonies regenerating on the selective medium were considered putative G418- or hygromycin B-resistant clones and transferred onto 1/10 SDA supplemented with 100 µg/mL G418 or hygromycin B.

### 2.6. Screening of the Desired Transformants

The desired transformants were finally screened by PCR and Southern blotting analyses. Total DNA was extracted according to a method described previously [18]. Aliquots of 50 to 100 ng of the total DNA were used as templates in the PCRs. For Southern blotting analysis, aliquots of approximately 10 µg of the total DNA were digested with an appropriate restriction enzyme, separated by electrophoresis on 0.8% (*w*/*v*) agarose gels, and transferred onto Hybond-N^+^ membranes (Cytiva, Tokyo, Japan). Southern hybridization was performed using an ECL^TM^ Direct Nucleic Acid Labeling and Detection System (Cytiva, Tokyo, Japan) according to the manufacturer’s instructions. All of the *T. benhamiae TbeMFS1*-lacking mutants and the revertant strains obtained were preserved as TIMM40016 (IFM 66350), TIMM40017 (IFM 66351), TIMM40018 (IFM 66832), and TIMM40019 (IFM 66833), respectively, in the culture collections of Teikyo University Institute of Medical Mycology (TIMM) and Medical Mycology Research Center, Chiba University (IFM), through the National Bio-Resource Project, Japan (http://www.nbrp.jp/ accessed on 8 July 2020).

### 2.7. Drug Susceptibility Testing of T. benhamiae TbeMFS1-Lacking Mutants

MICs were determined according to the broth microdilution method of the Clinical and Laboratory Standards Institute [19].

### 2.8. Search for Additional T. rubrum Potential CYH-Resistance Genes by Heterologous Expression in S. cerevisiae

A *T. rubrum* cDNA library [9], which was constructed in the pYES2-DEST52 vector, was used. *Saccharomyces cerevisiae* strains Y02409 were transformed with 1.0 μg of plasmid DNA, prepared from 3 × 10^7^
*E. coli* clones of the cDNA library. More than 10^6^
*URA3* transformants were obtained on MMD. Growing yeast cells were collected in batches and resuspended in deionized water at an OD of 1.0 (10^7^ yeast cells/mL). Then, 10^6^ yeast cells from this suspension were grown in 5 mL of MMG liquid medium to an OD of 1.0 (exponential phase) for induction of the *GAL1* promoter in pYES-DEST52. CYH- resistant clones were obtained by seeding 10^6^ growing cells onto plates of MMG and YPG medium containing 0.2 μg/mL CYH, a concentration equivalent to about 5 times the MIC of *S. cerevisiae* Y02409 for this antibiotic.

Plasmid DNA from individual CYH resistant *S. cerevisiae* transformants was isolated as previously described [20] and used to transform *E. coli* XL1 blue. The cloned cDNA inserts were identified by sequencing (Microsynth, Balgach, Switzerland).

### 2.9. Nucleotide Sequence Accession Numbers

The updated TruMFS1 cDNA sequence cloned in pYES2-DEST52 was submitted to GenBank with the identification number MZ182274. The updated protein sequences for both TruMFS1 and TbeMFS1 are available at the UniProtKB database (www.uniprot.org accessed on 2 June 2021) with the accession numbers F2SH39 and D4AXV8, respectively.

## 3. Results

### 3.1. MFS1 Characterization in T. rubrum and T. benhamiae

The cDNA encoding TruMFS1 was obtained from a *T. rubrum* cDNA library constructed with pYES2-DEST52 [9]. DNA sequencing revealed a 1.7 kb open reading frame encoding a polypeptide chain of 536 amino acids from the first methionine residue and corresponding to TERG_01623 from the whole *T. rubrum* genome sequence [21]. The analysis of the protein sequence showed that it corresponded to a transporter of the Major Facilitator Superfamily (MFS) [22] with 14 predicted transmembrane regions (Figure 2). The MFS transporters are single-polypeptide secondary carriers that are capable only of transporting small solutes in response to chemiosmotic ion gradients [23,24]. The closest homolog in *T. benhamiae* is ARB_01027 [25] but was annotated with an N-terminal extension. The new protein sequence for ARB_01027 without N-terminal extension shows 99.3% identity with TruMSF1/TERG_01623, named as TbeMFS1. TruMSF1 and TbeMFS1 differ by only two residues clustered in the N-terminal disordered region (Figure 2).

### 3.2. TruMFS1 Operates as a CHL Efflux Pump in S. cerevisiae

The ability of TruMFS1 to confer resistance to CYH was observed by heterologous expression of the cDNA hosted in pYES2-DEST52 downstream of the inducible *GAL1* promoter. This expression system was not adequate to test the ability of TruMFS1 to efflux compounds acting as inhibitors of mitochondrial translation such as CHL, ERY, and TET. Indeed, ATP production can be generated by glycolysis using fermentative carbon sources such as galactose, meaning that mitochondria are no longer necessary for fungal growth. To further examine the ability of MFS1 to efflux CHL, ERY, and TET, we needed to switch to a medium containing a non-fermentable carbon source that requires functional mitochondria for yeast growth. Therefore, we constructed the vector pMFS1-2 by introducing the gene encoding TruMFS1 in plasmid p426GPD, downstream of the strong *GPD* constitutive promoter (Table 1). *Saccharomyces cerevisiae* cells transformed with pMFS1-2 and the control plasmid p426GPD were tested for resistance to antibiotics by disc tests and serial dilutions on YCAG growth medium containing yeast extract casamino acids and glycerol.

*Saccharomyces cerevisiae* Y02409 transformed with p426GPD without *TruMFS1* cDNA showed sensitivity to CHL and ERY but not to TET on YCAG medium. Halos of inhibition around discs loaded with 20 and 100 µg of CHL are shown in Figure 3A.

Different results were obtained when *S. cerevisiae* was transformed with pMFS1-2 containing *TruMFS1* cDNA. Approximately one out of 50 CFU grew around the discs loaded with the same amounts of CHL (Figure 3A).

We also observed that approximately one CFU out of 50 pMFS1-2 transformants grew on a YCAG medium containing 100 µg/mL CHL using serial dilution tests (Figure 3B, line 2), while no colony was recorded with p426GPD transformants (Figure 3B, line 1). Yeasts from these growing colonies were then collected, resuspended in PBS, and deposited on a YCAG medium containing CHL. It appeared that all of the collected yeast cells could form colonies (Figure 3B, line 3). We also inoculated yeasts growing on YCAG with CHL onto MMD medium without CHL, and after two days of incubation, we reseeded the growing cells onto YCAG medium with CHL. Once again, about one CFU out of 50 pMFS1-2 transformants developed into colonies (Figure 3B, line 4). Thus, most cells lost their ability to form a colony resistant to CHL when they were no longer in the presence of this antibiotic. However, overall, our results suggest that the expression of *TruMFS1* conferred resistance of *S. cerevisiae* to CHL but not to ERY. With regard to the one yeast cell out of 50 generating a colony in a CHL medium with a non-fermentable carbon source after growth on a culture medium with dextrose, it is possible that the yeast must be in a certain growth phase to start growing. There was cell cycle heterogeneity of the yeasts when seeded on a culture medium in the presence of CHL.

### 3.3. MFS1 Plays a Major Role in Tolerance of T. benhamiae to CHL but Not to CYH

To examine the involvement of MFS1 in CHL resistance in dermatophytes, we generated mutants lacking the *MFS1 gene*, according to the gene replacement strategy shown in Figure 1B. To enhance the generation of such mutants, *T. benhamiae* was used as a recipient organism, for which more efficient genetic manipulation tools have been developed than for *T. rubrum*. The *T. benhamiae* Δ*Ku70* mutant ΔTbeKu70-5-1 without any selectable markers, which was generated by using the FLP recombinase-mediated site-specific recombination system [26], was used as a recipient strain for efficient homologous recombination. The transformation of ΔTbeKu70-5-1 with pAg1-*TbeMFS1*/T by *Agrobacterium tumefaciens*-mediated transformation produced many G418-resistant colonies on the selective medium, 25 of which were chosen at random, and analyzed using molecular biological methods. Finally, two transformants, ΔTbeMFS1-17-1 and ΔTbeMFS1-18-1, were confirmed to be the desired *MFS1* (*TbeMFS1*)-lacking mutants by Southern blot analysis (Figure 1C).

The growth properties of the obtained *T. benhamiae TbeMFS1*-lacking mutant ΔTbeMFS1-18-1 on solid medium containing CHL were compared with that of the parent strain ΔTbeKu70-5-1. The Δ*TbeMFS1* mutant grew well on SDA (Figure 4A) and SGA (Figure 4B) without CHL. In contrast, the growth of the Δ*TbeMFS1* mutant was repressed on SDA and SGA containing 200 µg/mL CHL. When the *TbeMFS1* gene was reintroduced into the same locus of the Δ*TbeMFS1* mutant ΔTbeMFS1-18-1, the growth of the obtained revertant strain (TbeMFS1/C-18-1-10) was restored in the presence of CHL (Figure 4A,B).

We measured the MICs of CHL and thiamphenicol (TAP), an analog of CHL, in the parent strain of *T. benhamiae* (ΔTbeKu70-5-1), two *TbeMFS1*-lacking mutants (ΔTbeMFS1-17-1 and ΔTbeMFS1-18-1) and two revertant strains (TbeMFS1/C-18-1-10 and TbeMFS1/C-18-1-16) by using the CLSI broth microdilution method. Both ΔTbeMFS1-17-1 and ΔTbeMFS1-18-1 mutants were 4-fold more susceptible to CHL than the parent strain ΔTbeKu70-5-1 and the revertant strains (Table 2). These results showed that the CHL tolerance of *T. benhamiae* was conferred by TbeMFS1. Similarly, both Δ*TbeMFS1* mutants were at least 8-fold more susceptible to TAP, compared with the parent strain and the revertant strains.

### 3.4. MFS1 Is Involved in Resistance to Azole Compounds but Not in CYH Resistance in T. benhamiae

We have previously shown that TruMFS1 was able to efflux azole compounds (FLC, ITC, VRC, KTC, and MCZ) and CYH when heterologously expressed in *S. cerevisiae*. To confirm its function as an azole transporter in *T. benhamiae*, we compared the azole sensitivities of the parent strain of *T. benhamiae*, Δ*TbeMFS1* mutants, and revertant strains (Table 2). The susceptibility to FLC and MCZ was significantly affected by the deletion, and the reintroduction of the *TbeMFS1* gene reversed this effect. The absence of significant observable effects with ITC and VRC could be explained by the action of the main transporters of these two compounds, MDR2 and MDR3, respectively, in *T. benhamiae* [9].

Surprisingly, Δ*TbeMFS1* mutants were not more sensitive to CYH, as compared with the parent strain (Table 2).

### 3.5. Search for Additional T. rubrum Potential CYH-Resistance Genes in S. cerevisiae

As the resistance of *T. benhamiae* to CYH could not be attributed to MFS1, we further explored the molecular basis of this resistance in dermatophytes by looking for other transporters conferring resistance in *S. cerevisiae* when overexpressed. Therefore, we took advantage of the existence of a previously constructed cDNA library [9], where *T. rubrum* cDNA sequences were cloned under the control of the *GAL1* promoter into the pYES2-DEST52 vector, to transform *S. cerevisiae* Y02409. Fifty CYH resistant clones were obtained as described in the Material and Methods. The plasmid DNAs of all clones were extracted and used to transform *E. coli* XL1 blue. Sequence analysis revealed that the ampicillin-resistant *E. coli* clones obtained from each *S. cerevisiae* transformant contained a 2.1–2.3 kb cDNA encoding TruMFS1. No other transcripts conferring CYH resistance were found.

## 4. Discussion

The origin of the dermatophytes corresponds to the early development of mammals [27]. Dermatophytes evolved from soil saprophyte fungi, which acquired the ability to efficiently degrade hard keratin into amino acids and short peptides in the process of recycling soil nitrogen. Many species gradually evolved to parasitize keratinous tissues of living animals in close contact with the soil. In this environment, dermatophytes were in contact with many microorganisms, which produced antibiotics and, at the same time, had the ability to resist them. In addition to confirming a role in resistance to antifungal azoles, we revealed that MFS1, an MFS-type transmembrane transporter, is also involved in resistance to CHL and TAP, but not to ERY. Together with CYH, CHL is widely used in laboratories to isolate dermatophytes from dermatological samples. While MFS1 does not play a significant role in CYH resistance in dermatophytes, we highlight the crucial role of this transporter in intrinsic CHL resistance. The suppression of MFS1 in *T. bemhamiae* strongly affected growth in the presence of CHL, which suggests a strong dependence on mitochondria of these specialized fungi. The aerobic route of energy production by the respiratory chain appears to be favored by dermatophytes. In this regard, it has previously been observed that these fungi stop growing and sporulate with a partial pressure of carbon dioxide [28] a means used to produce spores from low sporulating dermatophytes [29].

The results presented in Table 1 show that MFS1 plays an important role in the high FLC MIC of dermatophytes. The FLC MIC of *T. benhamiae* decreased by a factor of 4 to 8 when MFS1 was suppressed. FLC is a broad-spectrum fungistatic drug that has been shown to be effective in treating dermatophytosis [30,31]. Systemic ITC and FLC are considered second-line treatments for tinea unguium after systemic terbinafine [32]. FLC has a much higher MIC towards dermatophytes than any azole used in practice to treat dermatophytosis [33,34,35]. However, it is not always relevant to compare the efficacy of different drugs on the basis of their concentration, as many factors such as pharmacokinetics can play key roles in in vivo efficacy. Therefore, a higher MIC for FLC compared to other azoles does not necessarily indicate lower efficacy.

While many ABC-type transporters of fungal pathogens have been widely studied [9], much less is known about the roles of MFS-type transporters. Both TruMFS1 and TbeMFS1 are close to the *Aspergillus* aflT transporter. Despite its location within the aflatoxin gene cluster in *Aspergillus* species [36], aflT seems not to be involved in aflatoxin secretion and its substrates are not known so far [37]. Another MFS-type transporter with a similar structure, AaMFS54, has been shown to be required for resistance to fungicides, xenobiotics, and oxidants, as well as for full virulence, in *Alternaria alternata* [38].

Even though the heterologous expression of TruMFS1 in *S. cerevisiae* confers CYH resistance, the disruption of the *T. benhamiae* MFS1 encoding gene *TbeMFS1* did not alter the resistance of the fungus to this toxic compound. It is not the first time that the heterologous expression of a transporter in *S. cerevisiae* does not reflect the actual function of the same transporter in its host [39]. For instance, the gene *FLU1* in *Candida albicans* was identified as an FLC efflux pump via heterologous expression in *S. cerevisiae*, but the disruption of this gene in *Candida albicans* did not alter the resistance of the pathogen to this compound [40]. The lack of effect on the intrinsic resistance of *T. benhamiae* Δ*MFS1* mutants to CYH suggests the involvement of another mechanism of resistance. We considered *TruMDR2,* an ABC-type transporter that is 68.7% identical to *A. nidulans AtrD* for a role in the resistance of dermatophytes to CYH. *Aspergillus nidulans AtrD* acts as an efflux pump for CYH as well as nigericin and valinomycin in *A. nidulans* as Δ*AtrD* mutants display a hypersensitive phenotype to these compounds [41]. However, the deletion of *TruMDR2* in a *T. rubrum* clinical isolate overexpressing *TruMDR2* (TIMM20092) does not render the fungus more sensitive to CYH (data not shown), confirming previous results by Fachin et al. [11]. Moreover, heterologous *TruMDR2* overexpression in *S. cerevisiae* does not confer CYH resistance [9], and *T. rubrum* did not overexpress *TruMDR2* when challenged with cycloheximide [11]. On the other hand, the TruMDR1 transporter does not appear to be involved in CYH resistance either. Although overexpression of TruMDR1 was observed when the fungus was exposed to various toxic compounds including cycloheximide [10], overexpression of this gene in *S. cerevisiae* did not confer resistance to CYH [9]. Therefore, the main actor of CYH resistance in dermatophytes has yet to be identified. Still, with its capacity to efflux a wide range of toxic compounds such as azole compounds, CYH and CHL, MFS1 may help dermatophytes survive in the environment.

## Figures and Tables

**Figure 1 jof-07-00542-f001:**
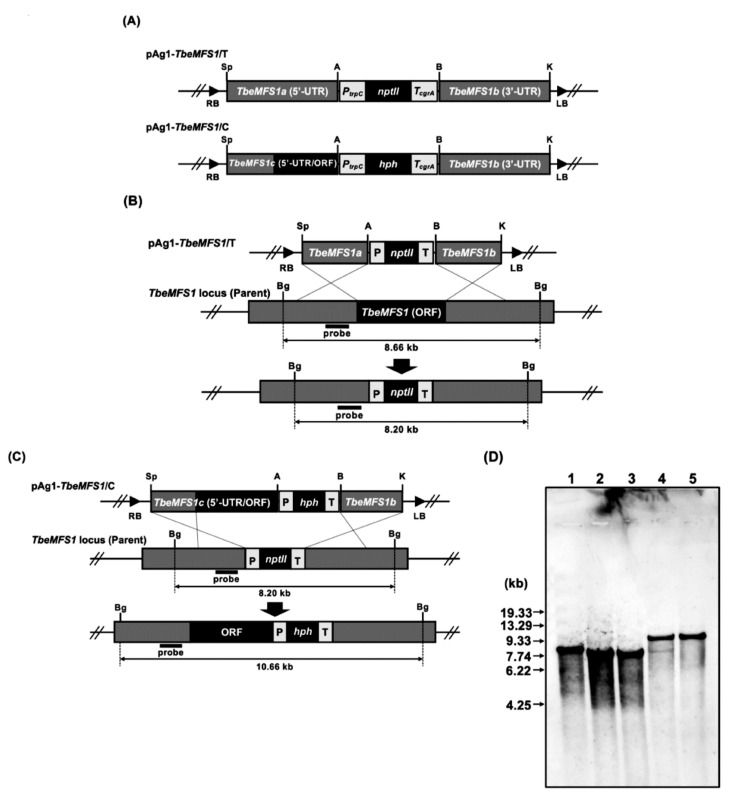
Disruption of the *TbeMFS1* gene of *T. benhamiae* ΔTbeKu70-5-1 by gene replacement strategy. (**A**) Schematic representation of the binary *TbeMFS1*-targeting and *TbeMFS1*-complementation vectors, pAg1-*TbeMFS1*/T and pAg1-*TbeMFS1*/C. The selectable marker cassettes are composed of *A. nidulans trpC* promoter (*PtrpC*), *E. coli nptII or hph* gene, and the *A. fumigatus cgrA* terminator (*TcgrA*). LB and RB, left and right borders, respectively; A, ApaI; B, BamHI; Bg, BglII; K, KpnI; S, SpeI. (**B**,**C**) Schematic representation of the *TbeMFS1* locus before and after homologous recombination with pAg1-*TbeMFS1*/T (**B**) and pAg1-*TbeMFS1*/C (**C**). (**D**) Southern blotting analysis. Lane 1, ΔTbeKu70-5-1 (Parent strain); Lane 2 to 3, ΔTbeMFS1-17-1, -and -18-1; Lane 4 to 5, TbeMFS1/C-18-1-10 and -16. An 878-bp fragment of the *TbeMFS1* locus was amplified by PCR with the primer pair AbF152-AbR141 (Appendix A) and used as a hybridization probe. DNA standard fragment sizes are shown on the left.

**Figure 2 jof-07-00542-f002:**
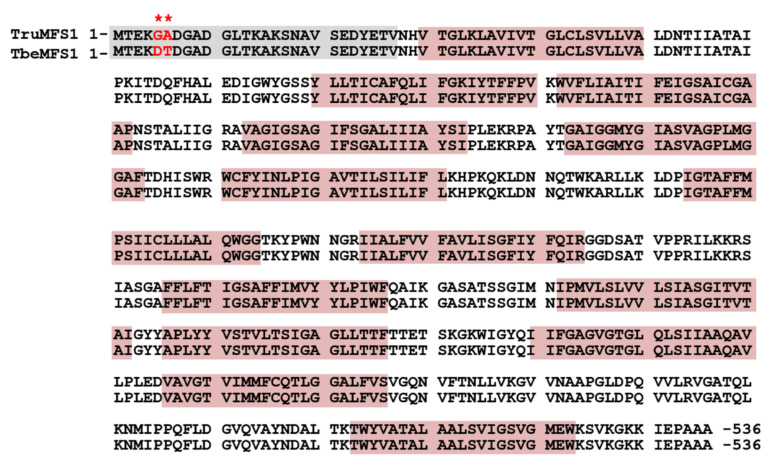
Alignment of the protein sequences of TruMFS1 (UniProt:F2SH39) and TbeMFS1 (UniProt:D4AXV8). The red stars indicate the residues that differ between the two polypeptides. The grey box shows the N-terminal disordered region. The red boxes indicate the 14 predicted transmembrane spans.

**Figure 3 jof-07-00542-f003:**
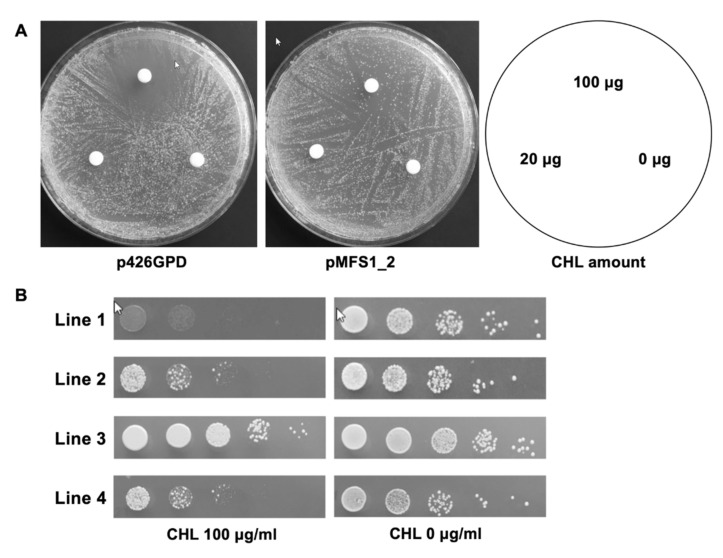
Resistance of *S. cerevisiae* overexpressing *TruMFS1* to CHL. (**A**): Disc tests: *S. cerevisiae* Y02409 p426GPD and pMFS1-2 transformants. (**B**): Serial dilution tests. Line 1: *S. cerevisiae* Y02409 p426GPD transformants. Line 2: *S. cerevisiae* pMFS1-2 transformants. Line 3: Yeasts from growing colonies in Line 2 were resuspended in PBS, and subsequently deposited on YCAG containing CHL. Line 4: Yeasts from the growing colonies of line 2 were cultured on MMD, then the yeast cells from generated colonies were inoculated on YCAG containing CHL.

**Figure 4 jof-07-00542-f004:**
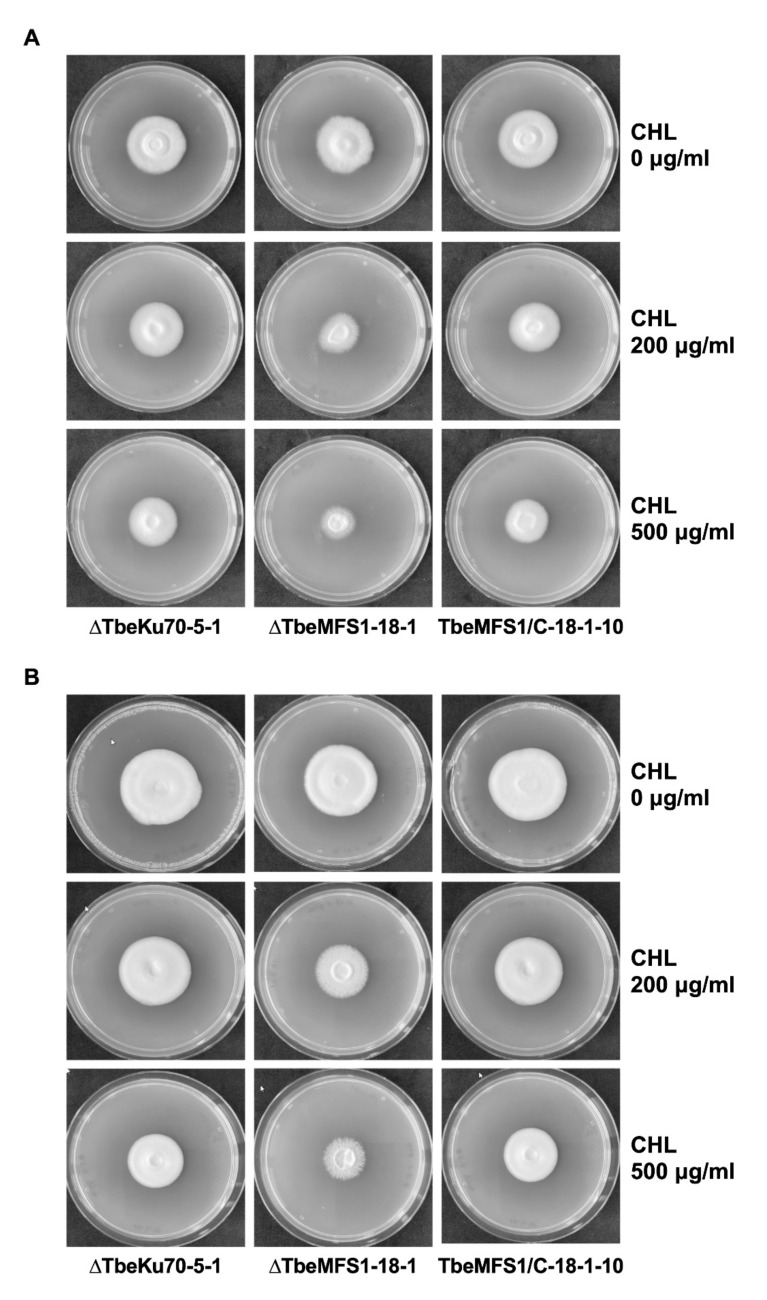
Targeted disruption of the *MFS1* gene makes *T. benhamiae* susceptible to CHL. The effect of CHL (200 and 500 µg/mL) in SDA medium (**A**) and SGA medium (**B**) on the growth of *T. benhamiae* ΔTbeKu70-5-1 (Parent strain), ΔTbeMFS1-18-1 (*TbeMFS1*-lacking mutant), and TbeMFS1/C-18-1-10 (Revertant strain).

**Table 1 jof-07-00542-t001:** Plasmids used in this study.

Plasmid	Description ^a^	Source or Reference
p426GPD	Yeast episomal plasmid (high copy number) carrying the 2 µ ori and containing the glyceraldehyde-3-phosphate dehydrogenase gene (GPD) promoter, flanked by a multiple cloning site and an XhoI-KpnI fragment of the cytochrome-c oxidase gene (CYC1) terminator, and the URA3 marker gene for the transformation of *S. cerevisiae*.	[16]
pMFS1	pYES2-DEST52 (Invitrogen) containing *TruMFS1* cDNA under the control of the *GAL1* promoter of *S. cerevisiae*.	[9]
pMFS1-2	p426GPD containing TruMFS1 cDNA flanked upstream by the GPD promoter and downstream by the CYC1 terminator.	This study
pAg1	Streamlined version of the binary vector pBIN19 containing sequences necessary for replication in *E. coli* and *A. tumefaciens* (*oriV* and *trfA*), *E. coli* neomycin phosphotransferase gene (*nptII*), and the transferable DNA (T-DNA) region, with a multiple cloning site within the T-DNA region	[17]
pSP72-PcFLP	pSP72 (Promega) containing the 5′-UTR of *Trichophyton mentagrophytes Ku80* (*TmKu80*) gene (GenBank accession no. AB427108) (1.5 kb), the *FLP* recombination target (5′-FRT), the promoter sequence of *A. nidulans* tryptophan C (*trpC*) gene (*PtrpC*) (GenBank accession no. X02390), *nptII*, the terminator sequence of *A. fumigatus cgrA* gene (*TcgrA*) (GenBank accession no. EAL84894), the promoter sequence of *T. rubrum* high affinity copper transporter (*ctr4*) gene (*Pctr4*) (TERG_01401), *Penicillium chrysogenum*-optimized FLP recombinase (*Pcflp*) gene, the terminator sequence of *Cryptococcus neoformans* phosphoribosyl anthranilate isomerase (*trp1*) gene (*Ttrp1*) (GenBank accession no. M74901), 3′-FRT and the 3′-UTR of *TmKu80* gene (1.5 kb).	[9]
pAg1-*hph*	*Cochliobolus heterostrophus* promoter 1 (GenBank accession no. M12304), *E. coli* hygromycin B phosphotransferase gene (*hph*), the terminator sequence of *A. nidulans trpC* gene (*TtrpC*)	[14]
pAg1-*TbeMFS1*/T	TbeMFS1a fragment (the 5′ UTR of *TbeMFS1* gene; 2.04 kb) (GenBank accession no. DAA75241), *PtrpC*, *nptII*, *TcgrA*, TbeMFS1b fragment (the 3′ UTR of *TbeMFS1* gene; 2.02 kb)	This study
pAg1-*TbeMFS1*/C	TbeMFS1c fragment (the 5′ UTR and ORF of *TbeMFS1* gene; 3.50 kb), *PtrpC*, *hph*, *TcgrA*, TbeMFS1b fragment	This study

^a^ ORF, open reading frame.

**Table 2 jof-07-00542-t002:** Susceptibility to antimicrobial agents of *T. benhamiae MFS1*-lacking mutants and revertants.

Species and Strains	Antimicrobial Agents (µg/mL)						
	CHL	TAP	CYH	FLC	ITC	VRC	MCZ
*Trichophyton benhamiae*							
ΔTbeKu70-5-1	1280	>2560	>4000	32	0.25	0.12	2
ΔTbeMFS1-17-1	320	160	>4000	4	0.25	0.06	0.5
ΔTbeMFS1-18-1	320	320	>4000	8	0.25	0.12	0.5
TbeMFS1/C-18-1-10	1280	>2560	>4000	32	0.25	0.12	2
TbeMFS1/C-18-1-16	1280	>2560	>4000	32	0.25	0.12	2
*T. interdigitale*							
ATCC MYA-4439	160	>2560	>4000	4	0.25	0.06	0.25

CHL, chloramphenicol; TAP, thiamphenicol; CYH, cycloheximide; FLC, fluconazole; ITC, itraconazole; VRC, voriconazole; MCZ, miconazole.

## Data Availability

The data presented in this study are archived in our laboratory and are available on request from the corresponding author (tsyamada@main.teikyo-u.ac.jp).

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
