# Peer review of "MFS1, a Pleiotropic Transporter in Dermatophytes That Plays a Key Role in Their Intrinsic Resistance to Chloramphenicol and Fluconazole"

_jof, 2021, doi:10.3390/jof7070542_

Round 1

Reviewer 1 Report

The authors investigated the roles of MFS1 in the intrinsic resistance of dermatophytes to cycloheximide and chloramphenicol, which are commonly used to isolate these fungi, and to azole antifungals.  The results have shown that MFS1 has a crucial role in the resistance of dermatophytes to chloramphenicol and fluconazole, but no to cycloheximide. This article presents new findings, the study seems generally well carried out, and it has potential clinical implications. Some aspects, however, could be improved.

Comments

 The drug efflux pumps to be activated depending on several factors, such as the type of drug, the exposure time and concentration of the drug, the strain/species of the fungus, and even the integrity of the other efflux pumps of the fungus. This characteristic may be a likely explanation for the fact that the heterologous expression of a transporter in S. cerevisiae does not reflect the actual function of the same transporter in its host.

Line 58: You are missing references. ...were also identified.

There are several other studies with efflux pumps in dermatophytes that have not been mentioned/cited.

Lines 76 and 178: Explain it better 1/10 SDA.

Line95: Please, clarify the statement "CHL, thiamphenicol and erythromycin (ERY) were dissolved in ethanol or H2O, respectively".

Line 286: Please, correct the word "reitant".

Line 312: Abbreviation for drugs is in the middle of the text.

Line 332: “No other transcripts conferring CYH resistance were found”. However, the pdr1 gene (TREG_02508) is overexpressed in the presence of cycloheximide (PMID: 16519017) and may be responsible for resistance to CYH.

Lines 375-383: Also, the deletion of TruMDR2 (TERG_08613) in T. rubrum did not overexpress TruMDR2 when challenged with cycloheximide and did not render the ΔTruMDR2 more sensitive to this drug (PMID: 16849730). These differences are worth discussing (see Comments above).

Author Response

Dear Reviewer 1

Firstly, we thank you for appropriate evaluation of our manuscript. Our responses to your requests and suggestions are as follows. Our work was modified in accordance with your requests and suggestions. All the corrections are highlighted by yellow in the revised manuscript file.

Sincerely,

Tsuyoshi Yamada

Reviewer 1

Line 58: You are missing references. ...were also identified.

There are several other studies with efflux pumps in dermatophytes that have not been mentioned/cited.

Response

References PMID16519017 (Reference No. 10) and PMID16849730 (Reference No.11) were added.

Lines 76 and 178: Explain it better 1/10 SDA.

Response

« [0.1% (w/v) Bacto peptone (BD Bioscience), 0.2% (w/v) dextrose, 2% (w/v) agar] » was added in the text.

Line95: Please, clarify the statement "CHL, thiamphenicol and erythromycin (ERY) were dissolved in ethanol or H2O, respectively".

Response

We changed the statement to as follows:

 « CHL (Sigma-Aldrich) and thiamphenicol (TAP) (Wako Pure Chemical) were dissolved in ethanol at a concentration of 20 mg/ml, whereas erythromycin (ERY) (Sigma-Aldrich) was dissolved in ethanol at a concentration of 10 mg/ml. »

Line 286: Please, correct the word "reitant".

Response

Done: “many G418-resistant…”

Line 312: Abbreviation for drugs is in the middle of the text.

Response

Abbreviations were introduced at line 54 and then used through the manuscript.

Line 332: “No other transcripts conferring CYH resistance were found”. However, the pdr1 gene (TREG_02508) is overexpressed in the presence of cycloheximide (PMID: 16519017) and may be responsible for resistance to CYH.

Response

Regarding this pertinent comment, we inserted the following statement into the "Discussion" section.

« On the other hand, the TruMDR1 transporter does not appear to be involved in CYH resistance either. Although overexpression of TruMDR1 was observed when the fungus was exposed to various toxic compounds including cycloheximide [10], overexpression of this gene in S. cerevisiae did not confer resistance to CYH [9]. Therefore, »

Lines 375-383:

T. rubrum did not overexpress TruMDR2 when challenged with cycloheximide and did not render the ΔTruMDR2 more sensitive to this drug (PMID: 16849730). These differences are worth discussing (see Comments above).

Response

« However, the deletion of TruMDR2 in a T. rubrum clinical isolate overexpressing TruMDR2 (TIMM20092) does not render the fungus more sensitive to CYH (data not shown), and heterologous TruMDR2 overexpression in S. cerevisiaedoes not confer CYH resistance [9]. »

was changed by the following sentences:

« However, the deletion of TruMDR2 in a T. rubrum clinical isolate overexpressing TruMDR2 (TIMM20092) does not render the fungus more sensitive to CYH (data not shown), confirming previous results by Fachin et al. [11]. Moreover, heterologous TruMDR2 overexpression in S. cerevisiae does not confer CYH resistance [9], and T.rubrum did not overexpress TruMDR2 when challenged with cycloheximide [11]. »

Reviewer 2 Report

In this paper the authors have evaluated the role of MFS1, an MFS-type transporter, in the intrinsic resistance of T. rubrum to chloramphenicol and cycloheximide. They used a heterologous expression of MFS1 in S. cerevisiae, and deletion mutants in T. benhamiae. Overall, it is shown that MFS1 is an efflux pump for chloramphenicol, and azoles in Dermatophytes but not for cycloheximide. Intrinsic resistance to cycloheximide is therefore probably mediated by another mechanism.

The aim of the study is of interest, as intrinsic resistance of Dermatophytes is not well understood and acquired resistance to different antifungal drugs, including azoles, is on the rise at a global level. The study is well designed, the methodology is appropriate, the conclusions are based on the results, and the paper is clearly presented.

In summary, the study has been well conducted and provide useful data.

I have only one minor comment:

  1. Line 357-358: This statement may be nuanced. It is always difficult to compare efficacy of different drugs on a µg/ml basis. There are many factors that play key role in in vivo efficacy such as pharmacokinetics. Therefore, higher MIC for FLC compared to other azoles does not necessarily indicate poorer efficacy.
  2. Line 353: typo, should read Table 2.

Author Response

Dear Reviewer 2

Firstly, we thank you for appropriate evaluation of our manuscript. Our responses to your requests and suggestions are as follows. Our work was modified in accordance with your requests and suggestions. All the corrections are highlighted by yellow in the revised manuscript file.

Sincerely,

Tsuyoshi Yamada

Reviewer 2

In summary, the study has been well conducted and provide useful data.

I have only one minor comment:

  1. Line 357-358: This statement may be nuanced. It is always difficult to compare efficacy of different drugs on a µg/ml basis. There are many factors that play key role in in vivo efficacy such as pharmacokinetics. Therefore, higher MIC for FLC compared to other azoles does not necessarily indicate poorer efficacy.

Response

We added the following statement to the end of the second paragraph of "Discussion section".

« However, it is not always relevant to compare efficacy of different drugs on the basis of their concentration, as many factors many factors such as pharmacokinetics can play key roles in in vivo efficacy. Therefore, a higher MIC for FLC compared to other azoles does not necessarily indicate lower efficacy. »

  1. Line 353: typo, should read Table 2.

Response

Corrected
